# Help Me Explore: Minimal Social Interventions for Graph-Based Autotelic Agents

## Abstract

In the quest for autonomous agents learning open-ended repertoires of skills, most works take a *Piagetian perspective*: learning trajectories are the results of interactions between developmental agents and their *physical environment*. The *Vygotskian perspective*, on the other hand, emphasizes the centrality of the *socio-cultural environment*: higher cognitive functions emerge from transmissions of socio-cultural processes internalized by the agent. This paper argues that both perspectives could be coupled within the learning of *autotelic agents* to foster their skill acquisition. To this end, we make two contributions: 1) a novel social interaction protocol called *Help Me Explore* (HME), where autotelic agents can benefit from both individual and socially guided exploration. In *social episodes*, a social partner suggests goals at the frontier of the learning agent's knowledge. In *autotelic episodes*, agents can either learn to master their own discovered goals or autonomously rehearse failed social goals; 2) GANGSTR, a graph-based autotelic agent for manipulation domains capable of decomposing goals into sequences of intermediate sub-goals. We show that when learning within HME, GANGSTR overcomes its individual learning limits by mastering the most complex configurations (e.g. stacks of 5 blocks) with only few social interventions.

## 1 Introduction

During early child development, open-ended play offers a learning context for exploring and mastering various skills. From solving puzzles to manipulating objects, children continuously and autonomously acquire complex behaviors driven by their intrinsic curiosity. Recently, artificial intelligence (AI) researchers have been interested in this open-ended learning framework to design artificial agents that can evolve in unbounded environments and grow repertoires of skills in a self-supervised fashion. Several argued that such artificial agents must be *autotelic*: they must be intrinsically motivated to represent, generate and pursue their own goals (Steels, 2004; Forestier et al., 2017; Colas et al., 2020b).

To this end, classic developmental approaches in AI relied on the agents' sensorimotor module to represent their goals (Eysenbach et al., 2018; Warde-Farley et al., 2018; Nair et al., 2018; 2020; Pong et al., 2019; Colas et al., 2019). This was inspired by the notion of *physical situatedness* in developmental psychology (Piaget, 1954; Needham & Libertus, 2011): children first learn to represent their goals through their embodied interactions with their surroundings, then use their intrinsic motivations to sample and pursue these goals. Unfortunately, this limits the behavioral diversity, as the distribution of *learnable* goals is highly grounded in the distribution of *previously encountered* goals. Hence, classic approaches in intrinsically motivated reinforcement learning (IMRL) usually rely on further assumptions, curricula and biases to drive the exploration process and increase behavioral diversity (Eysenbach et al., 2018; Warde-Farley et al., 2018; Nair et al., 2018; 2020; Pong et al., 2019; Colas et al., 2019).

Interestingly, developmental psychology has shown that children rely on two additional features: 1) *imaginative play* and 2) *social situatedness*. On the one hand, imaginative play allows children to imagine and target goals that they did not encounter previously; by substituting an object with another for example (Lewis et al., 1992; Stagnitti & Unsworth, 2000). The IMAGINE approach attempts to leverage this feature in IMRL by using language to imagine new goals (Colas et al., 2020a). On the other hand, social situatedness is essential for the emergence of sophisticated forms of in-

telligence (Vygotsky, 1994; Bruner, 1973; Tomasello, 2009). In the *Zone of Proximal Development* (ZPD) framework for instance (Vygotsky, 1978), a social partner (SP) boosts the learner's capabilities by first ensuring the mastery of simpler and foundational skills before the more complex ones (i.e. scaffolding). More generally, verbal and nonverbal interactions help acquire shared cognitive representations that make sens for both the SP and the learner. These findings have gained popularity in AI (Lindblom & Ziemke, 2002; Sigaud et al., 2021). In particular, we set ourselves in the framework of Sigaud et al. (2021), which investigates the design of artificial agents that can be both autotelic and social: autonomous teachable agents.

To facilitate these cultural interactions, research in cognitive science hypothesizes that human learning operates over structured and symbolic programs (Chater & Oaksford, 2013; Wang et al., 2019). These high-level representations offer a handful of efficient tools for manipulating information. Specifically, symbolic representations that are independent of languages and domains (such as images, symbolic predicates, ...) facilitate generalization of underlying concepts rather than memorization of specific situations. However, using these high-level tools requires a formal syntax and a cross-cultural understanding of their semantics. Fortunately, work in developmental psychology has shown that some of these requirements, precisely spatial relational predicates, are used by infants from a very young age (Mandler, 2012). This motivated recent works in developmental AI to endow artificial agents with such symbolic inductive biases (Tellex et al., 2011; Alomari et al., 2017; Kulick et al., 2013; Asai, 2019; Akakzia et al., 2020). Beyond facilitating the learning process of autotelic agents (Akakzia et al., 2020), this additional semantic layer should simplify social interactions.

**Contributions.** This paper makes steps towards the design of teachable autotelic agents. We argue that entangling social interventions within autotelic agents' individual learning facilitates the acquisition of a growing repertoire of skills. Furthermore, we show how social signals influence the learners' intrinsic motivations and help them break away from the constraints of their *physical situatedness*. To this end, our contributions are twofold:

First, we propose a novel social interaction protocol, **H**elp **M**e **E**xplore (HME), which intertwines social and autotelic episodes. In the former, a SP drives the agent towards its ZPD through goal proposals. They relies on a structured representation of the environment high-level goal space and a model of the agent's current exploration limits. Such a model is required for a good teacher to design lessons adapted to their students capabilities (Bruner, 1961). In the latter, the agent can either train to reach its known semantic goals or rehearse social proposals using an internalization mechanism.

Second, we introduce a GC-RL architecture named GANGSTR for **G**o**A**l-conditioned **N**eural **G**raph with **S**eman**T**ic **R**epresentations. GANGSTR is based on the Soft-Actor-Critic algorithm (SAC, Haarnoja et al., 2018), where both its actor and critic are implemented as Graph Neural Networks (GNNs, Scarselli et al., 2005; Gilmer et al., 2017; Battaglia et al., 2018). Furthermore, it gradually constructs a semantic graph of its discovered configurations and uses it to decompose goals into sequences of intermediate sub-goals.

We show that when engaged in the HME protocol, GANGSTR masters a diversity of complex goals it would not have discovered alone, spanning its entire goal space of around 70.000 semantic configurations. We also show that GANGSTR requires only light social interventions.

## 2 RELATED WORK

**Autotelic RL.** The quest for autotelic agents first emerged in the field of developmental robotics (Steels, 2004; Oudeyer & Kaplan, 2007; Nguyen & Oudeyer, 2012; 2014; Baranes & Oudeyer, 2013; Forestier & Oudeyer, 2016; Forestier et al., 2017) and recently converged with state-of-the-art deep RL research (Eysenbach et al., 2018; Warde-Farley et al., 2018; Nair et al., 2018; 2020; Pong et al., 2019; Colas et al., 2019; 2020a;b). Such agents rely on goal-conditioned RL algorithms (Schaul et al., 2015; Andrychowicz et al., 2017; Colas et al., 2020b) and Automatic Currciulum Learning (ACL) mechanisms (Portelas et al., 2020). Following the Piagetian tradition in psychology, most of these approaches consider individual agents interacting with their physical environment and leveraging forms of intrinsic motivations to power their exploration and learning progress (Bellemare et al., 2016; Achiam & Sastry, 2017; Nair et al., 2018; 2020; Burda et al., 2018; Pathak et al., 2019; Colas et al., 2019; Pong et al., 2019). These approaches sample goals from the set of already-known experiences, which limits their capacity to explore.

**Social Learning.** The ZPD describes the set of learning situations a child can solve with the help of a *more knowledgeable other* (Vygotsky, 1978). Throughout scaffolding and supportive activities, the SP uses their knowledge to drive learners beyond what they can do alone. Interactive learning research in AI investigates and models the ways human tutors can guide their agents' learning process (Thomaz & Breazeal, 2008; Argall et al., 2009; Celemin & Ruiz-del Solar, 2015; Najar et al., 2016; Christiano et al., 2017). A category of teaching signals requires the SP to point out sequences of actions and states using demonstrations or preferences (Argall et al., 2009; Christiano et al., 2017; Fournier et al., 2019). Another category assists the development of autotelic agents by providing them with challenging goals or guiding instructions (Grizou et al., 2013). As we want minimal interactions with the SP who does not show the agent how to act, we focus on the latter category.

**Semantic Goal Representations.** Developmental psychology suggests that humans rely on intuitive mechanisms of affordances, analogy and relational patterns to reason about their goals (Gibson, 1977; Holyoak, 2012). Endowing artificial agents with such inductive biases introduces a level of abstraction that is likely to facilitate the acquisition of complex tasks and the interaction with human teachers. Our set-up and semantic representation layer is similar to that of the DECSTR agent (Akakzia et al., 2020). However, there are several important discrepancies. First, in Akakzia et al. (2020), the training process was split into 3 sequential phases: sensorimotor learning, language grounding and instruction following. Given this split, instructions from the tutor could not be used to drive the learning process of the agent. By contrast, here sensorimotor learning and instruction following are intertwined, making it possible to focus on teaching curriculum questions. Besides, in Akakzia et al. (2020), the language grounding aspect was managed by a Language Goal Generator (LGG) component translating sentences from the tutor into semantic goals. Here, the language learning aspect is outside the scope of this study, as we consider that the tutor immediately communicates a goal in a semantic representation that the agent immediately understands.

**Graph-Based Exploration.** Humans solve novel problems by combining familiar skills and routines. They are able to represent complex systems within their environment as compositions of entities and their interactions (Navon, 1977; Marcus, 2001; Kemp & Tenenbaum, 2008). This motivates the design of artificial agents that learn structured representations of their world, such as graphs (Džeroski et al., 2001; Toyer et al., 2018). In this paper, we aim at endowing autotelic agents with a structured representation of their goal space. Close to us, the GO-EXPLORE algorithm uses a graph to represent a discrete archive of interesting states (Ecoffet et al., 2019; 2021). They map these states to discretized cells. They first reach a selected cell, based on heuristics such as recency or visit counts in memory, then explore from that state. We take a different approach based on the semantic character of the goals we consider. Using its goal space graph embedding, GANGSTR autonomously evaluates its performances when hopping from one goal to another. It then uses it as a proxy to determine the optimal path to reach complex semantic goal configurations.

## 3 METHODS

In this section, we introduce the HME interaction protocol (Section 3.1) and GANGSTR, a particular autotelic RL agent for manipulation domains. We first introduce the experimental environment and the underlying semantic goal space (Section 3.2). Then, we present GANGSTR's object-centered architecture, its internal semantic graph construction and decomposition processes (Section 3.3). Finally, we describe the application of HME to GANGSTR (Section 3.4). The anonymous code for the environment, agent and interaction setup as well as pretrained weights can be found at https://anonymous.4open.science/r/gangstr-2C4F.

### 3.1 THE HME SOCIAL INTERACTION PROTOCOL

The HME interaction protocol relies on *proposing social goals*. It involves an environment, a SP and an autotelic agent. First, the environment is goal-based and defined as an MDP. Second, the SP 1) is endowed with a structured representation of the environments' high-level goals and 2) has access to a model of the learning agent's current exploration limits. For high-level goals, we consider semantic predicates as an abstraction tool. For structured representations, we consider oracle graphs where nodes span all high-level goals. Two goals are linked if the SP considers them to be semantically adjacent. Third, the autotelic agent is able to 1) pursue high-level goals on its own and 2) remember the SP goal proposals.

Figure 1: Uniform Goal Sampling (up), ACL goal sampling (middle) and our proposed HME goal sampling (down). Histograms reflect the distribution of sampling probability for each method.

HME works as follows. Initially uninformed about its goal space, the agent performs random actions and unlocks easy goals. Once these goals are discovered, the interaction protocol is triggered. The agent can either 1) follow *autotelic episodes* where it uniformly samples a goal among the set of discovered goals or 2) follow *social episodes* where the SP proposes goals. During the latter, The SP uses their model of the agent's current exploration limits to first propose a *frontier goal* (discovered stepping stone). If the agent succeeds, the SP continues with an adjacent *beyond goal* (unknown goal). This allows the agent to explore and make progress in its ZPD. If it fails at reaching the beyond goal, it remembers the proposed pair so that it can autonomously rehearse social proposals during autotelic episodes. We refer to this rehearsal as *the internalization mechanism*.

Figure 1 illustrates the goal generation procedure in HME vs 1) the uniform case and 2) the ACL case, which do not use social interventions. In the first case, the goal sampler does not attend to a particular set within the discovered goals. In the second case, attention is given to goals with intermediate difficulty for the agent's current policy. These goals are at the frontier of the agent's current competence, but do not necessarily correspond to stepping stones towards new discoveries. Hence, both cases are constrained by the agents physical situatedness. By contrast, the social interventions and the internalization mechanism in HME help overcome this limit.

## 3.2 ENVIRONMENT AND SEMANTIC CONFIGURATIONS

We extend the *Fetch Manipulate* domain from Akakzia et al. (2020) to 5 blocks, itself a variant of the standard *Fetch* domains (Plappert et al., 2018). The agent embodies a Fetch robotic arm. It controls the 3D velocity of its gripper, the 1D velocity of its fingers (4D action space) and observes the Cartesian and angular positions and velocities of its gripper, fingers and of the 5 blocks it faces. In addition, it perceives a binary semantic representation of the scene similar to the one introduced in Akakzia et al. (2020). This representation asserts the presence (1) or absence (0) of the binary spatial relations *above* and *close* for each of the 10 pairs of objects. These 3 relations (*close*, *A above B*, *B above A*) applied to each of the 10 possible pairs lead to 30 binary dimensions (against $3 \times 3 = 9$ dimensions with 3 blocks, Akakzia et al., 2020). The resulting configuration space contains $2^{30}$ configurations, among which $\sim 70.000$ are physically reachable. These semantic representations are inspired by the work of Jean Mandler on a minimal set of spatial primitives children seem to be born with, or to develop early in life (Mandler, 2012).

In contrast with Akakzia et al. (2020), when resetting the setup, the 5 blocks are randomly placed in contact with the table. That is, any pair of blocks can be either far or close, but never stacked.

## 3.3 INTRINSICALLY MOTIVATED GOAL-CONDITIONED REINFORCEMENT LEARNING

This section introduces GANGSTR, an autotelic algorithm whose objective is to discover and master all configurations within the semantic configuration space. GANGSTR leverages the *Soft Actor-Critic* algorithm (SAC, Haarnoja et al., 2018) and *Hindsight Experience Replay* mechanisms (HER, Andrychowicz et al., 2017) to train a goal-conditioned policy and critic implemented by *Graph Neural Networks* (GNNs, Gilmer et al., 2017; Battaglia et al., 2018). In addition, the agent progressively constructs a graph of discovered configurations that is further used to decompose complex goals into sequences of easier ones. Let us now detail these components.

**Object-Centered Architecture.** A series of recent work argued for the importance of incorporating inductive biases in the architectures of neural function approximators (Battaglia et al., 2018; Gilmer et al., 2017; Zaheer et al., 2017; Kipf & Welling, 2016; Ding et al., 2020; Karch et al., 2020; Akakzia et al., 2020). In manipulation domains, agents need to handle separate objects, each characterized by a set of specific features. Here, what is important is the relations between objects, not their particular organisation in the agent's state vector. GNNs are particularly adapted to this case: they focus on relations between nodes in the graph, enforce permutation invariance and can handle sets of objects of different sizes.

Our implementation encodes the features of each object in a specific node and the binary semantic relations between object pairs in their edges. Further information about the agent's body (and actions for the critic) are encoded in the global features of the graph. Following Message Passing GNNs (MPGNNs, Gilmer et al., 2017), we conduct neighborhood aggregation and graph-level pooling schemes using a single network shared across nodes but separate for the actor and critic. These inductive biases facilitate the representation of spatial relations between objects and enforce a permutation invariance that boosts generalization, see details in Appendix A.2.

**Semantic Graph Construction.** As it interacts with the environment, the GANGSTR agent experiences new semantic configurations. This experience is used to progressively construct a graph of semantic configurations. This semantic graph should not be confused with the GNN's graph. Here, nodes correspond to semantic configurations and directed edges encode estimations of the agent's ability to transition from the initial to the final configuration. We call these measures success rates (SRs). After each episode and if they do not exist already, the agent creates two new nodes for the initial and final configuration and one edge for their transition (initialized to SR = 0.5). If the edge already exists, the agent updates the SR to reflect the episode's outcome (success or failure). For each edge in the graph, the SR is computed using an exponential moving average with $\alpha = 0.01$ that emphasizes the most recent outcomes. The estimated SR of a longer path can be computed by multiplying the estimated SRs of all its transitions.

**Construction Biases.** As training progresses, the number of edges in the semantic graph grows and can reach more than 500k edges at convergence. This results in very few outcomes stored in each edge, i.e. sample sizes too low to hope get reliable SR estimates. Unreliable SR estimates lead to poor choices of sub-goal decompositions which, in turn, leads to poor performance. To fight this issue, we implement two biases: **Attention Bias:** The number of edges in the graph can be drastically reduced if the agent only creates edges for transitions where the modified relations correspond to moving a single object. This can be seen as a focus on *small steps*, see Appendix A.3 for details; **Edge Permutations:** Since the GNNs of the GANGSTR agent enforce a permutation invariance which significantly boosts generalization across behaviors that are equivalent under the permutation of the objects' identities (permutation-wise equivalence), the SRs of permutation-wise equivalent transitions are expected to be close. To increase the sample size of our SR estimates, we share outcomes of all permutation-wise equivalent edges. This increases the sample size by a factor 120 (number of permutations of 5 elements). In turn, this results in $\sqrt{120} \approx 11$ times tighter confidence intervals for the SR estimations (Bernoulli parameter), see Appendix A.3 for details.

Note that these two biases rely on a single assumption that was already made in the design of the GNN: the agent knows which semantic features correspond to which object. Another approach could be to train a neural classifier to predict the binary outcome given the initial and final configurations and use its output preceding the sigmoid activation as a continuous estimate of SR. However, the two simple biases presented above were enough to reach good performance in our case.

**Automatic Decomposition.** At each individual episode, GANGSTR samples a goal uniformly from the set of discovered configurations. The agent then decomposes this goal into a sequence of easier, intermediate goals leading to it. This decomposition follows one of two strategies: **Shortest Path:** the agent selects one of the $k = 5$ shortest paths in terms of travelled edges; **Safest Path:** based on its SR estimates, the agent selects the path it is more confident in to successfully reach its goal. Always taking the shortest path can be risky when it includes unreliable transitions discovered in lucky trajectories. On the other hand, always taking the safest path can lead to long detours when SR estimates are unreliable. By taking one of the $k$ shortest paths, the agent explores short paths and collects outcomes to robustify the estimations of their SRs. At test time, the agent takes the safest path to maximize its probability to reach the goal configuration. In all cases, optimal paths are obtained with the Djikstra algorithm (Dijkstra, 1971). See Appendix A.5 for details.

## 3.4 HELPING GANGSTR EXPLORE WITH SOCIAL GOAL PROPOSALS

To apply HME to GANGSTR, we consider a simulated SP. The ratio of social vs autotelic episodes (see Section 3.1) is a hyper-parameter studied in our experiments. First, we use a similar structure as the one learned by the agent to encode the SP's domain knowledge of the semantic goal space (see Section 3.3). In practice, we use the network analysis library *networkit* to construct an oracle graph where nodes represent all imaginable semantic configurations within the 5-block manipulation domain. To model a semantically adjacent pair of configurations, we add a directed edge between two nodes whenever it is possible to reach one from the other by moving a single object. See Appendix A.4 for more details about oracle graph construction. Second, the SP models the agent's current exploration limits by keeping track of a list of all the configurations that it discovered so far. Thus, during social episodes, the SP can easily suggest frontier and beyond goals by simply projecting the elements of this list into its oracle graph. Finally, during autotelic episodes, GANGSTR can choose to either rehearse social proposals or uniformly select a goal that it discovered by itself. This is controlled by a hyper-parameter whose value is set to 0.5 in this work. When rehearsing social proposals, GANGSTR first considers a separate buffer where it has stored unsuccessful pairs of goals previously proposed by the SP. It then uniformly samples one and sequentially targets the corresponding frontier and beyond goals, exactly as if the SP was there.

## 4 EXPERIMENTS

This section studies the properties that emerge from applying HME to GANGSTR's autotelic learning process. In Section 4.1, we first vary the amount of social interventions to investigate how the presence of a SP affects the agent's performance and the underlying learning trajectories. We then study the role of the internalization mechanism during GANGSTR's training. In Section 4.2, we compare our methods to several ACL baselines and show that leveraging automatic goal generation curricula does not replace social interventions. In Section 4.3, we conduct ablative studies to highlight the importance of the learnt semantic graph. Additional studies and ablations are presented in Appendix A.6.

**Evaluation.** The subset of the $2^{30}$ expressible configurations the agent can physically reach is hard to compute *a priori*. To evaluate the agents, we hand-defined 11 classes of reachable configurations, each containing between 10 and 120 configurations (see Table 1 and Appendix A.6 for illustrations). This includes configurations where exactly $i$ pairs of blocks are considered *close* ($C_i$), configurations containing stacks of size $i$ ($S_i$), configurations containing pyramids of size 3 ($P_3$) and combinations of these. These classes are disjoint and their union does not cover the entire semantic configuration space (e.g. configurations $S_2\&C_2$ are not contained). They do contain, however, classes that we consider *interesting*; in the sense that they can be easily described by humans (e.g. $P_3\&S_2$ could be *"a tower of two and a small pyramid"*).

Evaluations are conducted each 50 cycles. During one cycle, the agent targets a goal, generates a path and follows up to 10 intermediate goals within rollouts of 40 timesteps. At test time, the agent is given 24 goal configurations uniformly sampled from each of the 11 classes (264 goals). A test episode is considered a success when the final configuration matches the goal configuration. The measure of the agent's success is the *global success rate*; the average of the 11 success rates per class. Testing episodes are conducted without exploration noise (deterministic policy) and are not added to the replay buffers (offline evaluations).

| Class | $C_1$ | $C_2$ | $C_3$ | $S_2$ | $S_3$ | $S_2\&S_2$ | $S_2\&S_3$ | $P_3$ | $P_3\&S_2$ | $S_4$ | $S_5$ |
|---|---|---|---|---|---|---|---|---|---|---|---|
| # Goals | 10 | 45 | 120 | 20 | 60 | 60 | 120 | 30 | 60 | 120 | 120 |

Table 1: Semantic classes used for evaluations. The $C_i$ class regroups all configurations where exactly $i$ pairs of blocks are *close*. The $S_i$ class contains all configurations where exactly $i$ blocks are piled up. The $P_i$ class contains all configurations where $i$ blocks form a pyramid. The & is a logical *AND*. All unspecified predicates are *false*, thus classes represent disjoint sets.

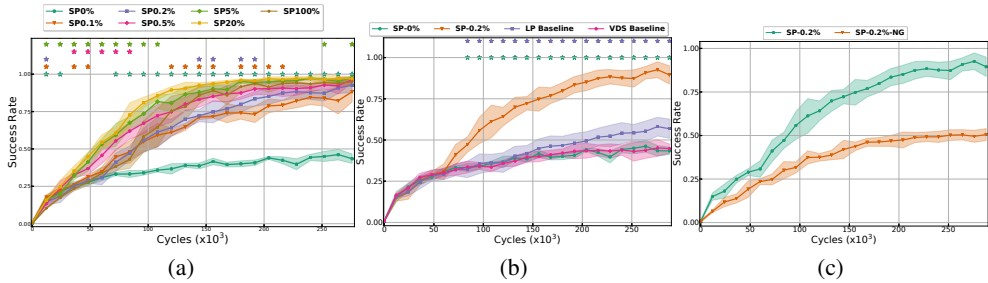

Figure 2: Global SR across training episodes for (a) various percentages of social episodes (Section 4.1), (b) ACL baselines (Section 4.2) and (c) the semantic graph ablation (Section 4.3). Mean ± standard deviation, computed over 5 seeds. Stars highlight statistical differences w.r.t. (a) SP-100%, (b) SP-0% (Welch's t-test with null hypothesis $\mathcal{H}_0$: no difference in the means, $\alpha = 0.05$).

## 4.1 HOW DOES HME AFFECT THE AGENTS' SKILL ACQUISITION?

**Global Performance.** Figure 2a presents the effect of various social interventions rates, from no social intervention (SP-0%) to exclusive social interventions (SP-100%). On the one hand, all social agents are found to significantly outperform the non-social agent SP-0% by more than 30%. Even 0.1% of social episodes is sufficient to trigger a significant difference in the learning trajectories. On the other hand, for enough social episodes, agents that combine autotelic learning and social learning show better sample efficiency than SP-100% as the global performance of SP-5% and SP-20% converges faster than SP-100%. This suggests that coupling autotelic and social learning in HME efficiently influences the agents' intrinsic motivations and helps them break away from their sensorimotor exploration constraints.

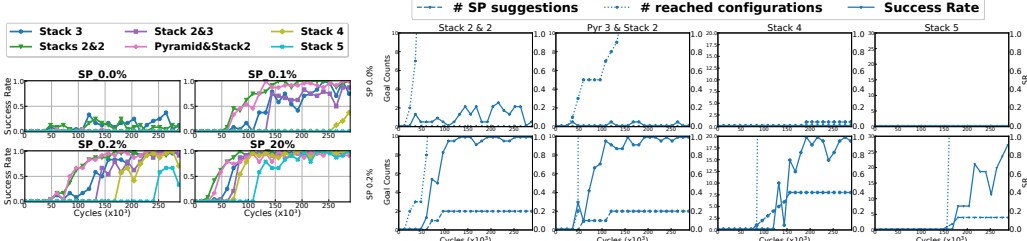

Figure 3: Success rates for each class with different ratios of SP interventions.

Figure 4: Learning trajectory of GANGSTR with 20% social intervention. We report the # of goals proposed by SP (dashed), the # of times GANGSTR encountered them (dotted) and its success rate (plain).

**Per-Class Performance.** Figure 3 shows the per-class performance of SP-0%, SP-0.1%, SP-0.2% and SP-20% agents. SP-0% fails to master complex configurations. However, adding only 0.1% of social interventions is enough to kick-off this mastery. With 0.2% of social interventions, all configuration classes are discovered, and reaching 20% efficiently allows to quickly master them. Here we only consider non-trivial evaluation classes for which non-social agents fail. Additional class-specific performance can be found in Appendix A.6.

**Induced Learning Trajectories.** Figure 4 presents the learning trajectories of SP-0% and SP-0.2% (rows) for several non-trivial classes (columns). These classes are considered hard because they either 1) are unlikely to be discovered from scratch with random exploration ($S_4$, $S_5$) or 2) can be discovered but are hard to reproduce ($S_2\&S_2$, $P_3\&S_2$). Plots of the remaining configuration classes are in Appendix A.6.

The learning trajectories of $S_4$ and $S_5$ show that GANGSTR does not discover these configurations unless the SP intervenes. For these classes, the dynamics follow the same pattern: 1) the agent discovers a stepping stone towards a yet-unknown configuration. This unknown configuration lies in the agent's ZPD, i.e. it is *just beyond* its current abilities. 2) The SP starts suggesting the stepping stone configuration as a goal and, if reached, immediately suggests the yet-unknown configuration from the ZPD (dashed line). 3) The agent discovers the configuration by exploring around the stepping

stone configuration suggested by SP. Enhanced by its object-centered architecture, it encounters it more and more often (dotted line). 4) Learning kicks in and the agent starts making progress on this new configuration; it will soon master it (plain line).

The learning trajectories of $S_2\&S_2$ and $P_3\&S_2$ show that a SP is not required for GANGSTR to discover them. However, although it managed to reach them on its own, it is unable to master them without the SP. In fact, the agents' performance on these classes only kicks off when social proposals start. This suggests that the role of the SP in HME is not only to help the agent discover previously unknown goals, but also to *facilitate* reaching them by showing the agent the easiest way via the stepping stone. This kind of intervention is minimal because the SP only sporadically suggests new directions for exploration. The internalization mechanism additionally removes burden from the SP (see next section).

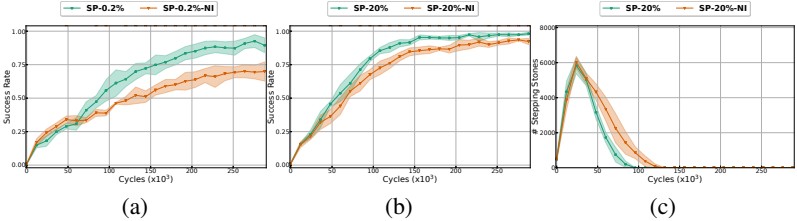

(a)  (b)  (c)

Figure 5: Global SR of (a) SP-0.2% and SP-0.2%-NI, (b) SP-20% and SP-20%-NI. (c) Counts of stepping stones during training for SP-20% and SP-20%-NI. Mean ± standard deviations, computed over 5 seeds. Stars indicate significant differences w.r.t. SP-X%. (Welch's t-test, $\alpha = 0.05$).

**Role of the Internalization Mechanism.** We denote by SP-0.2%-NI and SP-20%-NI GANGSTR agents receiving social interventions with an amount of respectively 0.2% and 20% but with no internalization mechanism. Figure 5a shows that with the internalization mechanism, SP-0.2% agents maximize their global SR while SP-20%-NI agents struggle (orange curve). Figure 5b shows that for higher amounts of social interventions, SP-20%-NI is able to maximize its global SR. Yet, the internalization mechanism in SP-20% makes it more sample efficient as it converges faster than SP-20%-NI (green vs orange curve). This reveals a specific influence of social interventions on the agent's learning process: including social goals removes the constraints on goal generation to what the agent previously encountered by itself in autotelic episodes (physical situatedness), even if the agent previously failed in the additional social episodes. Figure 5c highlights the number of stepping stones visited during training for SP-20% and SP-20%-NI. Since the number of stepping stones is finite, the role of the SP ends whenever there are no stepping stones left to propose, leaving GANGSTR to exclusively perform autotelic episodes. With the internalization mechanism in SP-20%, GANGSTR samples stepping stones more often and thus explores the goal space faster than SP-20%-NI. Hence, social episodes stop faster in SP-20% (at around 30% of the training budget), removing further burden from the SP.

## 4.2  CAN ACL METHODS REPLACE SOCIAL INTERVENTIONS?

In this section, we confront HME to ACL methods. Both influence goal selection during training episodes of autotelic agents: in ACL, more focus is given to the discovered goals that are neither too hard nor too easy, that is the goals where the learning progress (LP) of the agent is high, while in HME, the focus is given to the stepping stones that let agents discover goals that are beyond their individual training capabilities. To this end, we introduce two ACL baselines: 1)**LP Baseline:** We consider 11 initially empty buckets, each corresponding to an evaluation class. This baseline uses domain knowledge: whenever the agent discovers new goals, it is able to associate it with the corresponding bucket. During training, the agent first selects a bucket according to its LP. Then it uniformly selects a goal within the chosen bucket. 2)**VDS Baseline:** We train 3 Q-value networks to evaluate the difficulty of goals. During training, at the beginning of each episode, the agent samples 1000 goals in the list of discovered goals, evaluates their Q-values, computes a score reflecting the uncertainty about each goal. Applying softmax to the obtained scores yields a probability distribution that the goal sampler uses to more often select goals presenting higher uncertainty. When updating their networks, both baselines use their buffers to perform uniform experience replay. Additional details are presented in Appendix A.6.

Figure 2b presents the global SR of the ACL baselines, SP-0.2% SP-0%. First, the VDS Baseline performs on par with SP-0%, which relies on simple uniform selection of goals during training. This suggests that the goals for which the value functions present high uncertainty which are more sampled in the VDS Baseline do not necessarily correspond to the stepping stones that could potentially help GANGSTR discover new goals. Second, the LP Baseline outperforms SP-0%. This is not surprising since it uses additional domain knowledge. However, the it is still unable to span the entire goal space and maximize its SR. Finally, SP-0.2% outperforms both ACL baselines, suggesting that although ACL methods focus in principle on sampling the goals that present the highest learning progress, they are still unable to break away from the constraints of physical situatedness. This implies that these methods could not replace the social interventions introduced in HME.

## 4.3 ABLATION STUDY

In this section, we present an ablation study to assess the relative importance of the semantic graph in GANGSTR. A discussion about the importance of the GNN-based architecture is presented in Appendix A.6.

We denote by SP-0.2%-NG the ablation of SP-0.2% that does not use a semantic graph. Instead of generating a sequence of intermediate sub-goals, it directly conditions its policies on the final goal. Figure 2c presents the global SR of SP-0.2%-NG and SP-0.2%. It shows that SP-0.2% outperforms SP-0.2%-NG by more that 45%. The reason behind the failure of SP-0.2%-NG is that many semantic configurations, especially those that require a long sequence of manipulations to be reached, are hard to achieve without the decomposition tool. This issue is further enhanced by the sparsity of the reward: the rewarding signal struggles to propagate to earlier states. By contrast, SP-0.2% uses its graph-based decomposition to generate *checkpoints* that help smooth the reward signal. A more in-depth study of the effect of the graph-based decomposition is presented in Appendix A.6.

## 5 CONCLUSION AND FUTURE WORK

This paper contributes HME, a new interaction protocol involving a SP and an agent learning goal-oriented behaviors. HME couples autotelic and social learning. Endowed with a structured representation of the goal space and a model of the agent's exploration limits, the SP catalyzes the development of the learner by first proposing stepping stones and then continuing with adjacent beyond goals. The latter represents the agent's ZPD: the set of goals it cannot reach unless social guidance is provided. When it fails in reaching a beyond goal, the agent can remember it, as well as the stepping stone unlocking it. An internalization mechanism in HME then allows the agent to break away from its sensorimotor constraints by rehearsing these failed social goals during autotelic episodes. To demonstrate the benefits of HME, we present GANGSTR, an autotelic agent for manipulation domains endowed with an object-centered architecture and a graph-based representation of its semantic goal space. With the latter, GANGSTR can decompose its goals into sequences of intermediate sub-goals.

HME yields *light social interventions*. It removes from the SP the burden of providing demonstrations and directly manipulating the environment. Besides, whenever the agent has experienced all stepping stones, the SP stops intervening and the agent continues learning on its own. Our results show that GANGSTR requires only 0.2% of HME's social episodes to master all possible semantic configurations and that with 20% of social episodes, the role of SP ends at 30% of the training time. We believe social interventions could be further reduced. For example, we could combine ACL methods to HME for more efficient autotelic goal sampling. Besides, we could further optimize social interventions by implementing an automatic competence-based query mechanism from the learner.

The future of teachable autonomous agents lies in the diversity of social interactions they could learn from. Future work will aim at integrating multiple sources of social guidance using the framework of *pragmatic frames* introduced by Bruner (Bruner, 1985) and further refined in the context of robotics (Vollmer et al., 2016; Rohlfing et al., 2016). Pragmatic frames are verbal or non-verbal patterns of goal-oriented behaviors that evolve over repeated interactions between a learner and a teacher. As teachers and learners co-evolve these frames and learners learn to identify them from context, learners can integrate multiple sources of guidance in a way that will feel more natural for humans.

BROADER IMPACT STATEMENT

We believe human intervention is crucial in the quest for more explainable and safer autonomous robots. We exhibit a social protocol where our RL agents can have minimal interactions with a simulated SP and grow their repertoires of skills. Thus, this paper contributes in facilitating human teaching of artificial robots. By releasing our code and providing clear explanations of the number of seeds, error bars and statistical testing of our results, we believe we help efforts in reproducible science and allow the wider community to build upon and extend our work in the future.

ACKNOWLEDGMENTS

Anonymized for submission

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

# A APPENDIX

## A.1 PSEUDO CODE

Algorithms 1 and 2 present the high-level pseudo-code for the autotelic and social learning episodes.

| **Algorithm 1** Autotelic Learning | **Algorithm 2** Social Learning |
|---|---|
| 1: **Require** Env $E$, | 1: **Require** Env $E$, social partner $SP$ |
| 2: Initialize policy $\Pi$, semantic graph $\mathcal{G}_s$, path estimator $PE$, buffer $B$, internalization buffer $B_i$. | 2: Initialize policy $\Pi$, semantic graph $\mathcal{G}_s$, path estimator $PE$, buffer $B$, internalization buffer $B_i$. |
| 3: **loop** | 3: **loop** |
| 4:     goals = [] | 4:     $g \leftarrow SP$.propose_frontier($\mathcal{G}_s$) |
| 5:     **if** rehearse social episodes **then** | 5:     $path \leftarrow PE$.sample_path($g, \mathcal{G}_s$) |
| 6:         $(g, g_b) \leftarrow B_i$.sample() | 6:     **loop** $g_i \in path$ |
| 7:         goals.append($g$) | 7:         $trajectory \leftarrow E$.rollout($g_i$) |
| 8:         goals.append($g_b$) | 8:         $\mathcal{G}_s$.update($trajectory$) |
| 9:     **else** | 9:         $PE$.update($trajectory$) |
| 10:         $g \leftarrow \mathcal{G}_s$.sample_node() | 10:         $B$.update($trajectory$) |
| 11:         goals.append($g$) | 11:     **if** $g$ is reached **then** |
| 12:     **loop** $g \in$ goals | 12:         $g_b \leftarrow SP$.propose_beyond($\mathcal{G}_s, g$) |
| 13:         $path \leftarrow PE$.sample_path($g, \mathcal{G}_s$) | 13:         $path \leftarrow PE$.sample_path($g_b, \mathcal{G}_s$) |
| 14:         **loop** $g_i \in path$ | 14:         **loop** $g_i \in path$ |
| 15:             $trajectory \leftarrow E$.rollout($g_i$) | 15:             $trajectory \leftarrow E$.rollout($g_i$) |
| 16:             $\mathcal{G}_s$.update($trajectory$) | 16:             $\mathcal{G}_s$.update($trajectory$) |
| 17:             $PE$.update($trajectory$) | 17:             $PE$.update($trajectory$) |
| 18:             $B$.update($trajectory$) | 18:             $B$.update($trajectory$) |
| 19:         $\Pi$.update($B$) | 19:     **if** $g_b$ is not reached **then** |
| 20: **return** $\Pi, PE, \mathcal{G}_s$ | 20:         $B_i$.store($g, g_b$) |
| 21: | 21:     $\Pi$.update($B$) |
| 22: | 22: **return** $\Pi, PE, \mathcal{G}_s$ |

## A.2 OBJECT-CENTERED ARCHITECTURE

In the proposed *Fetch Manipulate* environment with 5 blocks, the GANGSTR agent perceives:

1. the low-level geometric states. Since the 5 objects share the same attributes dimensions (positions, velocities, orientations), the behavior with respect to an object should be independent from the object's attributes.

2. the high-level semantic configurations. Since the relations between all the pairs of objects share the same predicates (*close*, *above*), the behavior with respect to a binary semantic relation should be independent from the considered pair.

Thus, it is natural to encode both object-centered and relational inductive biases in our architecture. To do so, we model both the agents policies and critics as MPGNNs. We consider a graph of 5 nodes, each representing a single object. All the nodes are interconnected, yielding a compact graph of 20 directed edges. Furthermore, we consider the agent's body attributes as global features of the policy networks and both the agent's body attributes and the actions as global features of the critic networks. A single forward pass through this graph consists in three steps:

1. **Message computation** is performed for each edge. The features of the considered edge are concatenated with the features of the edge's source and target nodes before being fed to a first shared neural network $NN_{\text{edge}}$.

2. **Node-wise aggregation** is performed for each node. The features of the considered node are concatenated with an aggregation of the updated features of all the incoming edges. The resulting vectors are then concatenated with the global features of the graph before being fed to a second shared neural network $NN_{\text{node}}$.

3. **Graph-wise aggregation** is performed once for all the graph. The updated features of all the nodes of the graph are aggregated and fed to a third neural network $NN_{\text{readout}}$.

The aggregating function needs to be permutation invariant. We use max pooling for the *node-wise aggregation* and summation for the *graph-wised aggregation*. The final output of $NN_{\text{readout}}$ is either the action (in the case of the actor) or the $Q$-value (in the case of the critic).

### A.3    HIGH-LEVEL GRAPH-BASED POLICY CONSTRUCTION

**Attention Bias -** When exploring its environment, the agent is likely to encounter transitions that are complex to reproduce (for example while pushing one object using another). Taking these transitions into account while constructing the graph would affect the quality of the agent's high-level representation, leading to the generation of long and risky intermediate path based on inaccurate estimates of the SR. To circumvent this issue, we assume that the agent only considers *small steps* when constructing an edge between two nodes. Borrowed from the intuition in human multi-object manipulation as humans tend to manipulate one object at a time, we consider a step to be small whenever the changed predicates between the initial and final one correspond exactly to one object. This is easy to determine in practice thanks to the definition of our semantic configurations: they are binary vectors where each dimension corresponds to the evaluation of a predicate (close or above) on a certain pair of objects. Hence, looking at the dimensions that changed during a step allows to determine what predicates changed, and thus verify if these predicates are relative to one or more objects.

**Edge Permutation Bias -** To estimate the success rates of each transition in the graph, we share outcomes across transitions that are equivalent by permutations of the objects' identities (see Section 3.3). Figure 6 illustrates this permutation bias.

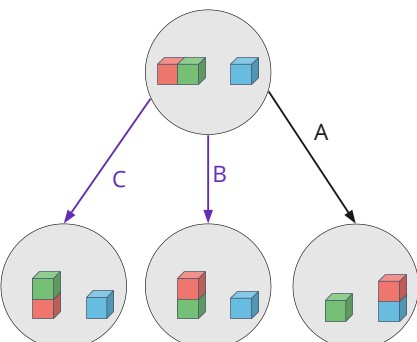

Figure 6: Estimates of SR in edges C and B can be shared.

**Automatic Decomposition** GANGSTR uses its semantic graph to decompose its goals into sequences of intermediate sub-goals. This semantic graph is gradually constructed during training: nodes get added whenever new semantic configurations get discovered, and directed edges get created between two nodes whenever it is possible to reach one from the other by moving only one object. The value initially allocated to each edge is 0.5. Each time the agent targets a goal, it keeps track of its corresponding successes and failures. This allows the agent to update the value of each edge using EMA. Hence, an edge between two nodes reflects how certain is the agent of reaching one from the other. The two methods that are combined by GANGSTR to sample its path are the following:

1. **Shortest Path:** the agent selects one of the $k = 5$ shortest paths in terms of travelled edges.

2. **Safest Path:** the agent selects the path which maximizes the SR, that is

$$path^* = \arg\max_{path} \prod_{edge \in path} \text{SR}(edge) = \underset{path}{\text{argmin}} \sum_{edge \in path} -log(\text{SR}(edge)).$$

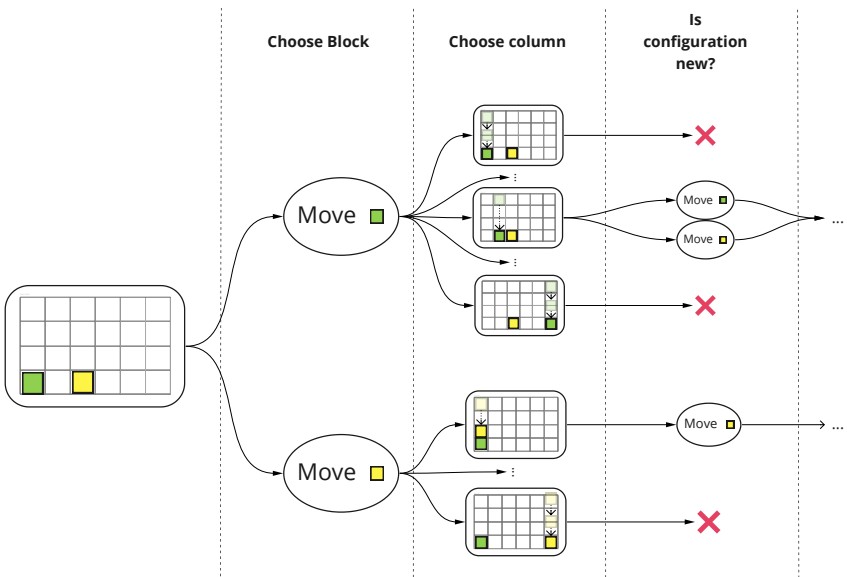

Figure 7: Simplified example with 2 blocks in a 2D grid of the oracle tree construction procedure

## A.4 REPRESENTING THE SOCIAL PARTNER'S KNOWLEDGE

Similar to the agent's, we choose to represent the SP's domain knowledge as a directed semantic graph. This facilitates determining stepping stones in the agent's capabilities. However, it is impractical to manually enlist all the imaginable configurations and decide whether a pair should be linked or not in a graph. Semantic configurations can be infeasible for two reasons: 1) they are semantically impossible to obtain—e.g. two objects cannot both be above each other 2) they are physically impossible to achieve—e.g. in the case of inverted pyramids.

To avoid enlisting all semantic and physical constraints, we design a 3D grid within which each cell can contain one object. Initially, all the blocks are initialized in different columns of the grid so that they are all far from each other (*root* node). From this state, we can select one object and move it to another column. If that column already contains another object, than the first one will be stacked above the second. By doing one action at each step, we can extract the current semantic configuration and link it to the previous one in the oracle graph. Iteratively repeating this process yields a tree starting from the *root* node. See Figure 7 for a 2D illustration of the described process for two blocks. Note that this engineered process only serves to evaluate the capacities of the agent and is not used by the agent itself in any way.

## A.5 IMPLEMENTATION DETAILS

This part includes details necessary to reproduce results. An anonymous under construction version of our code is available at https://anonymous.4open.science/r/gangstr-2C4F.

GNN-*based networks.* Our object-centered architecture uses two shared networks, $NN_{\text{edge}}$ and $NN_{\text{node}}$, respectively for the message computation and node-wise aggregation. Both are 1-hidden-layer networks of hidden size 256. Taking the output dimension to be equal to $3\times$ the input dimension for the shared networks showed better results. All networks use ReLU activations and the Xavier initialization. We use Adam optimizers, with learning rates $10^{-3}$. The list of hyperparameters is provided in Table 2.

*Parallel implementation of* SAC-HER. Our experiments rely on a *Message Passing Interface* (Dalcin et al., 2011) to exploit multiple processors. Each of the 24 parallel workers maintains its own replay buffer of size $10^6$ and performs its own updates. Updates are summed over the 24 actors and the updated actor and critic networks are broadcast to all workers. Each worker alternates between 10 episodes of data collection and 30 updates with batch size 256. To form an epoch, this cycle is repeated 50 times and followed by the offline evaluation of the agent.

Table 2: Hyperparameters used in GANGSTR.

| Hyperparam. | Description | Values. |
|---|---|---|
| $nb\_mpis$ | Number of workers | 24 |
| $nb\_cycles$ | Number of repeated cycles per epoch | 50 |
| $nb\_rollouts\_per\_mpi$ | Number of rollouts per worker | 10 |
| $rollouts\_length$ | Number of episode steps per rollout | 40 |
| $nb\_updates$ | Number of updates per cycle | 30 |
| $replay\_strategy$ | HER replay strategy | $final$ |
| $k\_replay$ | Ratio of HER data to data from normal experience | 4 |
| $batch\_size$ | Size of the batch during updates | 256 |
| $\gamma$ | Discount factor to model uncertainty about future decisions | 0.99 |
| $\tau$ | Polyak coefficient for target critics smoothing | 0.95 |
| $lr\_actor$ | Actor learning rate | $10^{-3}$ |
| $lr\_critic$ | Critic learning rate | $10^{-3}$ |
| $\alpha$ | Entropy coefficient used in SAC | 0.2 |
| $\alpha_{EMA}$ | EMA coefficient for SR edge estimation | 0.01 |
| $edge\_prior$ | Default value for edges' SR | 0.5 |
| $shortest_paths$ | Number of shortest paths to sample from | 5 |
| $shortest\_safest\_ratio$ | Ratio of alternation between shortest and safest paths | 0.5 |
| $internalization\_prob$ | The probability of rehearsing the SP's goal proposals | 0.5 |

## A.6 ADDITIONAL RESULTS

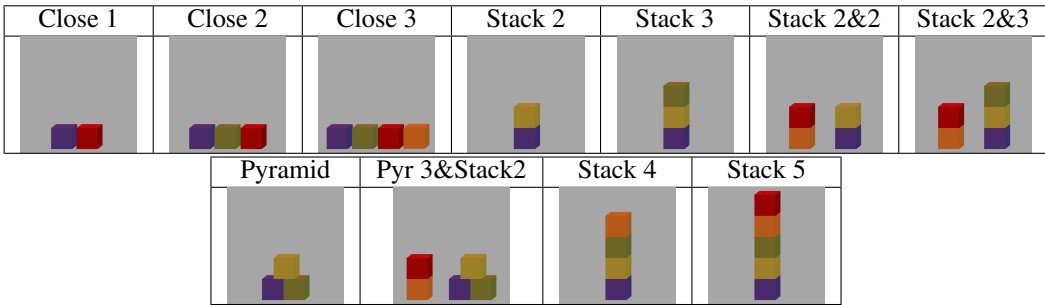

Table 3: The different semantic classes used in evaluation. The class *Close i* regroups all semantic configurations where $i$ pairs of blocks are close.

**Evaluation Classes.** Table 3 illustrates the 11 evaluation classes presented in the main paper. For the sake of simplicity here, we only represent the blocks that are concerned by the underlying predicates. All the predicates associated with the other blocks are values to 0. We use a hard-coded function to generate a random configuration given the identifier of the considered class. We also use a dictionary where keys are configurations and values are identifiers of the classes to keep counts of either the SP proposed goals or the agent's encountered and achieved goals.

**Additional Learning Trajectories.** In Figure 8, we present additional learning trajectories of a particular GANGSTR agent SP-0.2%. We can see that for some classes, social intervention (dashed lines) precedes goal reaching (dotted lines). These classes are considered hard for the agent to discover on its own (for example Stack 4, Stack 5). Other classes do not require social intervention to be discovered since the agent's own random exploration is sufficient (for example Close 3, Stack 2, Pyramid). Once configurations are reached, the agent is able to sample them when updating its policy and learn how to achieve them when targeted (plain lines).

**Addition Class-wise Performance** Figure 9 depicts the success rates per class of semantic goal configurations for 3 GANGSTR agents (rows) with different ratios of social intervention (columns). These additional results follow the claims we made in the main paper. In fact, when no social intervention is provided (SP-0%), the agents only masters the goals that it is able to discovered through random explorations. This excludes goals that are hard to find, namely high stacks and

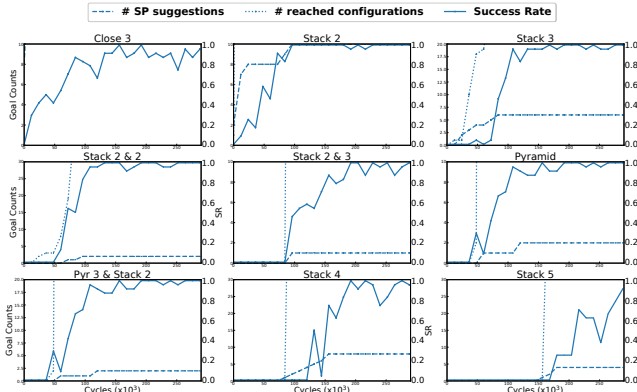

Figure 8: Learning Trajectory of GANGSTR with social intervention ratio of 0.2%. We report the number of goals proposed by SP (dashed), the number of times the goals were encountered (dotted) and the success rate (plain).

compositions of classes. Increasing the ratio of social intervention (i.e looking from left to right in Figure 9 allows to effectively grow the agent's repertoire of skills.

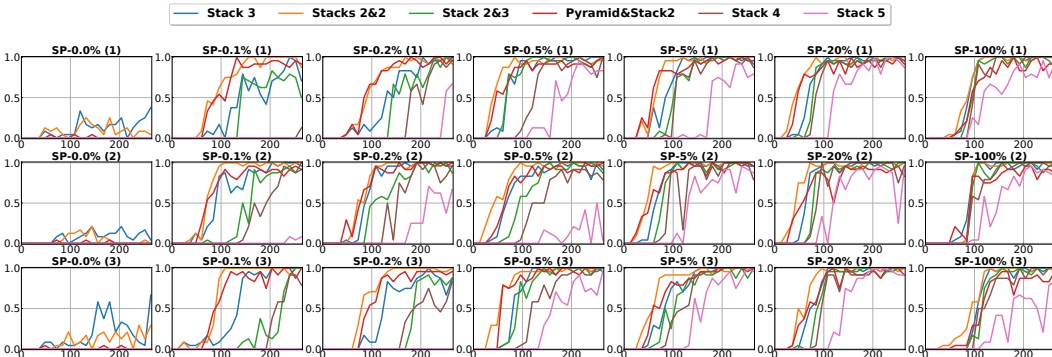

Figure 9: Success Rates per class of semantic goals configurations for different percentages of social episodes. The Performance of 3 agents is give nor each value.

**Decomposition Path Study.** GANGSTR uses its semantic graph to decompose goals into sequences of intermediate sub-goals. In the main paper, the length of the path (i.e. the number of intermediate sub-goals) is not constrained. Here, we study the impact of this parameter on GANGSTR's skill acquisition. For a percentage of social episodes fixed at 0.2%, we introduce the agents SP-0.2%-L, where $L$ corresponds to the maximum length of the generated path. We consider $L \in [2, 3, 4]$. Note that $L = 2$ exactly corresponds to the ablation of the semantic graph introduced in the main paper as SP-0.2%-NG (For $L = 2$, the only considered configurations are the initial and final ones). In practice, we take the entire generated path, consider the initial configuration and the last $L - 1$ configurations, including the target goal. Comparing to other approaches such as forward recall (considering the first components of the path) or uniform recall (uniformly selecting elements within the path) is left for future work. Figure 10 presents the global performance metrics for SP-0.2%, SP-0.2%-2, SP-0.2%-3 and SP-0.2%-4. It shows that SP-0.2%-3 outperforms SP-0.2%-2 without semantic graph. That is, considering only the penultimate configuration as an intermediate sub-goal is sufficient to increase the performance by 25%, since some complex goals that SP-0.2%-2 was unable to master are now unlocked using the unique intermediate goal. However, $L = 3$ is not enough to maximize the global SR. Considering SP-0.2%-4 further increases the global SR. Actually, it performs on par with SP-0.2% on average while introducing more variance. This suggests that with higher values of $L$, GANGSTR is more precise when targeting final goals, and that the estimation of the edge's values (GANGSTR's SR from a configuration to another) gets more accurate as training progresses.

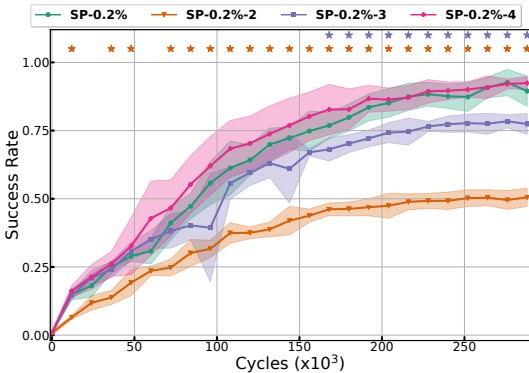

Figure 10: Global SR across training episodes for different values of the decomposition length. Mean $\pm$ standard deviation, computed over 5 seeds. Stars highlight statistical differences w.r.t. SP-0.2% (Welch's t-test with null hypothesis $\mathcal{H}_0$: no difference in the means, $\alpha = 0.05$).

**Amount of Social Interactions: Stepping Stones Counts.** In HME, social episodes automatically stop whenever the number of stepping stones is annealed (the SP has no more goals to propose). This leaves the agent to exclusive autotelic episodes, relying both on its physically-discovered goals and its internalization mechanism. To study the amount of social interactions needed, we introduce Figure 11, presenting the evolution of the counts of stepping stones for SP-5%, SP-0.5%, SP-0.2% and SP-0.1%. The curves show that for higher percentages of social interactions, the stepping stones are faster annealed, resulting in an early stopping of the SP interventions. This proves that social interventions could be further optimized: if the agent owns an efficient autonomous goal sampling module (based on its competence for example), the SP could intervene with a percentage that is 1) high enough to faster anneal stepping stones but 2) still low to remove their burden.

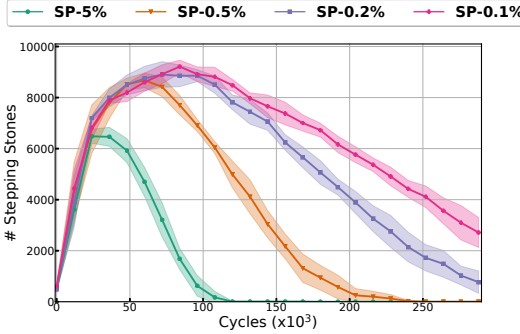

Figure 11: Counts of stepping stones during training for several percentages of social episodes.

**GNN Ablation Discussion.** Previous work investigated the benefits of object-based architectures compared to fully-connected networks. More specifically, we refer to the DECSTR agent where the same object manipulation environment was used. Similar to us, DECSTR uses the SSACRL algorithm. However, it leverages Deep-Sets to model both the actor and the critic. To ensure generalization and transfer of skills between pairs of objects, DECSTR uses a shared network that takes features from each pair as well target and current semantic goals as input. Then, outputs from all pairs are aggregated before being fed to a readout function. For performance to kick off, DECSTR necessarily requires to automatically leverage a curriculum during its training. However, its applicability was still limited to the case of 3-object manipulation environment. By contrast, we use MPGNNs instead of Deep-Sets to model the actor and critics in GANGSTR. Furthermore, no curriculum learning is required for the performance to kick off. In fact, through the simple uniform sampling of training goals, GANGSTR is able to master a large set of diverse semantic goal configuration in 5-object environments, including Stacks of 3 or higher and compositions of constructions. This suggests that, by opposition to Deep-Sets that look at pair of objects separately, the message passing function in GANGSTR allows all the information relative to all the pairs to reach every module. Hence, object-

centered architecture that rely on message passing are more suited to deal with semantic goals that involve spatial binary semantic predicates.

