# OpenReview forum: "Help Me Explore: Minimal Social Interventions for Graph-Based Autotelic Agents"
_ICLR.cc/2022/Conference — ICLR 2022 Submitted_

### Official Review · Reviewer_Pnco · 2021-10-25

**Correctness:** 3
**Technical Novelty And Significance:** 3
**Empirical Novelty And Significance:** 3
**Recommendation:** 5
**Confidence:** 3

**Main Review:**


1, I'm confused about the semantic knowledge graph construction. Which part of graph features that SR measures? The knowledge graph should be considered as structured knowledge where it depicts the real activities/knowledges. It seems that it exists some transition probabilities that related with the semantic graph, which shouldn't be considered as KG.

2, It is not clear in A.3 that how to alleviate the attention bias in small steps. How to quantify the small compared to others? What's the performance?

3, There should be more ablation studies over automatic decompositions. E.g. why the k=5 is the shortest path?

**Summary Of The Paper:**

This paper has explored the area of teachable autonomous agents where it highlights two contributions a design of the social interaction protocol and a learning architecture called GANSTER. Generally, the extensive experimental results have demonstrated the effectiveness of proposed method. However, there are still a few concerns. Attached below


**Summary Of The Review:**

Generally, this paper propose an innovative research question in teachable autotelic agents. However, some aspects of the structures, especially the semantic knowledge graph constructions.

---

> ### Author Response · Authors · 2021-11-17
> **Answer to Reviewer 3 (2/2)**
>
> __Concerns about ablation studies__
>
> R3 seems to be concerned about ablation studies, asking why k=5 is the shortest path and suggesting further investigations concerning decomposition. We believe there has been a misinterpretation of the parameter k here. In fact, k does not represent the length of the path here, but rather the number of sampled short paths from which the agent samples one. Namely, if k=1, the agent only samples the shortest path. If k=5, the agent selects the 5 shortest paths from the initial to the final configurations, then uniformly selects one of these 5 paths. We find that, in practice, this fosters exploration and results in overcoming possible edge estimation errors that may occur if only one path had been considered.
> That being said, there were no constraints on the length of the paths sampled by the agent using its semantic graph: given an initial configuration and a final configuration, the paths are constructed by considering all the intermediate sub-goals.
> In the new version of the paper, we conduct an additional study where we constrain the maximum length of a sampled path and study its effect on the agent's behavior.
>
>
> __Conclusion__
>
> We believe that the main concern of R3 is related to the denomination of our agent's semantic graph as a Knowledge Graph and to the attention bias. In the new version of the manuscript, we remove all the occurrences including KG and replace it with "the agent's semantic graph". We think this removes all ambiguities that arise from using the KG qualification. We also give more details about the attention bias mechanism, which we hope provide sufficient clarification to R3.

---

> ### Author Response · Authors · 2021-11-17
> **Answer to Reviewer 3 (1/2)**
>
> We thank Reviewer 3 for their helpful feedback to improve the manuscript. Here, we answer comments that were specific to Reviewer 3. More general concerns about the structure of our paper are addressed in detail in the general response.
>
> __Concerns about the semantic graph__
> Reviewer 3 (R3) seems to be mostly concerned about the fact that we call the semantic graph of goals used by our agent a “Knowledge Graph” (KG). From our understanding of R3’s comment, a KG is not supposed to have transition probabilities, which is the case of our semantic graph. We rechecked several definitions of KGs (https://www.ibm.com/cloud/learn/knowledge-graph,http://ai.stanford.edu/blog/introduction-to-knowledge-graphs/). Indeed, calling the semantic graph that our agent uses a KG is ambiguous. We have changed this denomination in the new version of the manuscript. We thank R3 for pointing that out.
>
> Here we clarify how our agent's semantic graph is used: first of all, this graph contains the semantic goal configurations that the agent __knows__. Initially empty, it uses rollouts conducted by the agent in the environment to create 1) a node modeling the __starting goal configuration__, 2) a node modeling the __achieved goal configuration__ and 3) a directed __edge linking both nodes__. The existence of an edge between two nodes of the agent's semantic graph means that the target node __can be__ reached directly from the source node by moving exactly one object. The value of an edge indicates how well the agent can perform the corresponding transition. In practice, we keep track of the successes/failures encountered when performing the corresponding transition and use Exponential Moving Average (EMA) to estimate the Success Rate (SR) corresponding to that transition (that is, the probability that the agent succeeds to reach node j starting from node i). These estimation values are later used during intermediate sub-goals sampling to determine the more appropriate path that an agent could follow to reach a final goal given its current competence (i.e. its current policy).
>
> We believe that the new framing of the paper removes the ambiguity as we now omit the term KG and give more focus to how SR estimations are used to determine candidate paths leading to a final goal.
>
> __Concerns about the attention bias__
>
> R3 raises questions about the attention bias and how to quantify the small steps when constructing an edge between two nodes. We thank R3 for this question as it helped us reconsider our description of the attention bias in A.3. Here we give further details about this: a semantic goal configuration is a binary vector where each dimension corresponds to the evaluation of a predicate on a certain pair of objects (e.g. the first dimension corresponds to close(__object1__, object2), the second dimension to close(__object1__, object3), ..., each one is equal to 1 if and only if the corresponding predicate is verified). Hence, achieving a semantic goal comes down to fulfilling a set of constraints in the current scene (exactly 30 constraints, 10 for the “close” predicate and 20 for the “above” predicate). When considering a rollout to be used to update the graph (i.e. to construct 2 nodes and a directed edge linking them as described above), we compare the achieved goal configuration to the starting goal configuration: if the only dimensions/predicates that changed correspond to exactly one object (for example only the values of dimensions close(__object1__, object2) and close(__object1__, object3) has changed), we allow the corresponding edge to be created. This is exactly what the attention bias does. It is borrowed from the intuition in human multi-object manipulation as humans tend to manipulate one object at a time. In practice, adding this bias allows to limit irrelevant edge creation, which ruins the agent's path generation (especially when the policy is still random, the agent can move an object using another object instead of its gripper).
>
> In the new version of the paper, more attention was given to appendix A.3 to better explain the use of the attention bias.

---

### Official Review · Reviewer_a8j4 · 2021-11-02

**Correctness:** 2
**Technical Novelty And Significance:** 2
**Empirical Novelty And Significance:** 2
**Recommendation:** 3
**Confidence:** 4

**Main Review:**


This is an interesting paper, however, as it stands, its contribution is quite limited.

The authors also refer to literature about learning that is tangentially related to the model they use. At the end, it seems that the model itself is quite simplified and the model is only loosely connected to the concepts that are presented in the first part of this paper. It seems that the model itself is essentially a mixture of conditioned goal-oriented learning and social learning. It seems to the reviewer that the term "autotelic" might remind the reader to autonomous generation of goals, which does not seem the case for this work.

In any case, the most problematic aspect is related to the actual evaluation of the approach. In fact, the evaluation is purely experimental. This is perfectly fine, but the authors consider a rather specific example and it is rather hard to generalize for it. In particular, the part about social learning and curriculum learning appear to be effective for the task considered in the paper, but this might not be the case in general. The reviewer would suggest the authors to consider either another set of tasks and/or a more in-depth discussion of the theoretical foundations of the approach. The reviewer understands that it might not be possible in general, but, at the same time, I believe that the authors might provide the readers with a more in-depth analysis of the contribution of the self-directed learning and social learning.

Another aspect that it would be interesting to analyze in more detail is the amount of social learning that is needed and how that should be structured in order to apply the approach to other situations and tasks.

The authors should also try to improve the presentation of the proposed solution by separating the discussion of the proposed solution and the application to their specific task. For example, the authors present the problem of task decomposition, but that really refers to the problem at hand.

I found the discussion of goal-conditioned learning quite interesting, but again, it seems to me that the authors go from the general solution to their specific problem without clearly introducing it (and making clear that the problem under consideration has some specific characteristics and constraints).

The discussion of the social learning part (Section 3.3) is rather high-level (the references to the psychology literature also appear slightly out of place here). The reviewer would like to suggest adding a more detailed description of the actual social learning mechanisms.


**Summary Of The Paper:**

The paper discusses a model for learning a task (5-block manipulation) using both "self-directed" (which the authors define as "autotelic" learning and social learning. I would say that the contribution is quite modest since the authors demonstrates the validity of their approach only through simulations of a specific game. It is difficult to evaluate the generality of the proposed approach.

**Summary Of The Review:**

The authors consider a solution of a task (5-block manipulation) using "self-directed"/goal-conditioned learning and social learning. According to the reviewer, the contribution of this work is limited, since it is difficult to understand if the solution generalizes to other tasks. In fact, the evaluation is based only on that specific task.

---

> ### Author Response · Authors · 2021-11-17
> **Answer to Reviewer 2 (2/2)**
>
> __Discussed literature is tangentially related to the model__
>
> R2 seems to be concerned about the positioning of our paper as they qualify the literature that we introduced as only "tangential" to our model. We thank R2 for this feedback, but we somewhat disagree here. Our paper is positioned within the developmental AI literature, which takes clear inspiration from the field of developmental psychology and how children, who are autotelic learners by nature, acquire diverse skills. In humans, social interventions such as proposing goals help learners overcome their individual learning limits. Our proposed HME protocol exhibits these properties, and our GANGSTR algorithm is a proof-of-concept in the object manipulation domain. We agree that considering other social learning mechanisms is interesting. However, rigorously describing them is out of our scope, especially given the page limits. In fact, we are particularly interested in reducing the burden of the SP and we believe that social goal proposals are a good candidate for that purpose. We added a discussion in the conclusion of the new version of the manuscript to highlight this.
>
>
> __More in-depth analysis of the amount of social learning needed__
>
> R2 expresses their concern about the amount of social learning needed in our approach. We thank R2 for suggesting an in-depth study of this aspect. In the new framing of the paper, we introduced additional results which better quantify the needed amount of social learning. Since social interventions in our HME protocol correspond only to goal proposals, we study the evolution of the number of stepping stones —that is, goals that the agent knows and that are directly connected to undiscovered goals yet. We believe this provides an additional quantification of the amount of social learning needed to cover all possible stepping stones, spanning the whole learnable semantic goal space.
>
> __Conclusion__
>
> We think the main concerns of R2 were related to the insufficient elicitation of our contributions and the evaluation of our approach. In the new framing of the paper, we clearly separate our main conceptual contribution from our second algorithmic one. Due to time and the number of pages limit, we could not provide results on additional environments. However, we have chosen to provide more in-depth studies in a rich enough multi-goal environment rather than more superficial studies in many environments.

---

> > ### Comment · Reviewer_a8j4 · 2021-11-28
> > **Assessment after Receiving Authors' Replies**
> >
> > The reviewer would like to thank the authors for taking care of replying to my review so thoroughly. However, I have still concerns about this work.
> >
> > In particular, my concerns in terms of actual novelty (and potential impact) of this paper are still there. Above all, again, my worry is that the purely experimental evaluation of this work is insufficient to prove itself validity even if I acknowledge that it was extended in this revised version. The presentation has indeed improved, but the actual contributions made by the authors are still of rather limited interest and they are not evaluated in a convincing general way in my opinion.

---

> > > ### Author Response · Authors · 2021-11-29
> > > **Answer to Reviewer a8j4 after receiving feedback about the revised version**
> > >
> > > The authors would like to thank the reviewer for taking the time to submit an additional comment. We understand the general feeling expressed by the reviewer, but stated as is, the review does not provide much guidance on how to improve our work.
> > >
> > > First, the reviewer questions "actual novelty" without pointing to any third party papers. We would be very interested in including such references in our future work. Could the reviewer indicate papers that make our work not novel enough to be published as is?
> > >
> > > Second, the reviewer questions “potential impact”. Impact is something discovered late after the publication of a paper. It is hard to foresee from one person, as many different researchers have many different interests (see e.g. the positive feedback from Reviewer SdCn).
> > >
> > > Then the reviewer questions whether a purely experimental evaluation is sufficient to prove validity of the work. Could the reviewer point to any inconsistency in our experimental methods that would devalue the validity of our work? We believe that rejecting our paper because the results are only experimental and we do not provide a theoretical analysis does not conform with ICLR 2022 reviewing guidelines, that we remind below (the highlight is ours):
> > >
> > > “Answer three key questions for yourself, to make a recommendation to Accept or Reject:
> > >
> > > 1) What is the specific question and/or problem tackled by the paper?
> > >
> > > 2) Is the approach well motivated, including being well-placed in the literature?
> > >
> > > 3) Does the paper support the claims? This includes determining if results, **whether theoretical or empirical**, are correct and if they are scientifically rigorous.”
> > >
> > > If after reading this rebuttal and the ICLR reviewing guidelines, the concerns of the reviewer remain, would it be possible to express them in a way that helps us pinpoint what could be done to improve our paper?
> > >
> > > We would like to thank again the reviewer for their time.

---

> ### Author Response · Authors · 2021-11-17
> **Answer to Reviewer 2 (1/2)**
>
> We thank Reviewer 2 for their helpful feedback to improve the manuscript. Here, we answer comments that were specific to Reviewer 2. More general concerns about the structure of our paper are addressed in detail in the general response.
>
> __Comments on contributions and presentation of the paper__
>
> Reviewer 2 (R2) seems to be mostly concerned about the contributions of our paper. We thank R2 for their constructive comments and suggestions on that aspect. These helped us properly restructure the focus of our paper. Namely, the summary advanced by R2 shows that our work was understood as exclusively dedicated to a quite specific 5-block manipulation task. However, our main contribution is conceptual. It consists of the Help Me Explore (HME) interaction protocol, that combines learning from intrinsically-motivated goal generation and learning from goals proposed by a social partner. HME benefits from both the physical situatedness (Piaget, 1954) and the social situatedness (Vygostky, 1978) of autonomous agents. The paper argues that interesting properties emerge from the HME protocol given efficiently sufficient conditions: 1) an environment stated in the form of a MDP; 2) a social partner (SP) with a structured representation of the underlying high-level goal space; 3) an autotelic artificial agent.
>
> We believe the new framing of the paper puts more focus on this main contribution and clearly disentangles it from our second contribution —the GANGSTR autotelic agent dedicated to manipulation domains. This better highlights the interesting properties in GANGSTR that emerge from the HME architecture, namely: 1) intrinsic motivations break away from physical situatedness thanks to our new __internalization module__ and 2) social interventions are _light_, as they require no demonstrations nor direct manipulation of the environment and only occur occasionally (at small percentages of the training time, actually 0.2% is sufficient given the new framing of the paper). Besides, the new framing of the paper implies that social goal proposals stop when there are no more stepping stones for the SP to propose, leaving the agent to exclusively individual episodes for the rest of the training time.
>
> __Concerns about evaluation__
>
> R2 raises an important question about the evaluation of our methods, suggesting that additional tasks should be considered and that we should provide a more in-depth discussion of the theoretical foundations of the approach. We agree with R2 that experiments in more diverse domains would give further insights on the effectiveness of our methods. This is true, but in the trade-off between in-depth studies of a single benchmark and more shallow studies of a variety of benchmarks, we prefered targeting depth and considered the object manipulation domain as sufficiently representative. In fact, our 5-object manipulation environment and the considered relational predicates yield a large goal space of tens of thousands of possibly reachable goal configurations, each of different complexity. Specifically, a large set of these goals (for example stacks of 4 or higher and combination of constructions) are not reachable when using stochastic policies and inducing exploration noise. We believe that, with the new reframing of our manuscript and the separation of the main conceptual contribution from the second algorithmic one, we could better explain that block manipulation is only a proof-of-concepts domain to show how the HME protocol is efficient in helping autotelic agents to better explore their goal space.
>
> __The model itself is essentially a mixture of conditioned goal-oriented learning and social learning__
>
> R2 seems to be concerned about the nature of our HME architecture and the use of the term "autotelic". We thank R2 for raising this issue as it led us to rethink our exposition of the different components of the HME protocol. We believe the new framing of the paper, which we introduce in detail in the general response, and especially the use of a new internalization mechanism, removes this ambiguity. Indeed, the previous version of the way we presented the architecture could be understood as a __decoupled__ mixture of autotelic learning and social learning. In autotelic learning, the agent pursues goals according to its own intrinsic motivations, while in social learning it pursues goals proposed by the SP. Allowing the autotelic agent to internalize the SP goals does better, as the presence of a social partner __influences autotelic agents' intrinsic motivations__: now, the agent is not only able to sample goals that it encountered, but can also rehearse goals that were proposed by SP __but are still not physically encountered__. This clearly helps the agent break away from physical situatedness.
>
> We believe this introduces an additional novelty to our work. In fact, to our knowledge, we are the first to study the effect of social interventions on the intrinsic motivations of autotelic agents.

---

### Official Review · Reviewer_SdCn · 2021-11-03

**Correctness:** 4
**Technical Novelty And Significance:** 3
**Empirical Novelty And Significance:** 3
**Recommendation:** 8
**Confidence:** 3

**Main Review:**

There are few points that I appreciate in his paper:

•	The paper investigates an important problem in robotics and AI in general, namely learning with guidance from a social partner. It relies on previous developmental psychology studies (e.g., the notion of Zone of Proximal Development by Vygotsky), and it proposes novelty in terms of a hybrid agent that both explores configurations for his own goals by interacting with its physical environment, as well as getting guiding instructions from a social partner (a social learning component).

•	The paper suggests an interesting approach for intervention by the social partner: it first let the agent to discover configurations on its own, including configurations at the boundaries of its current abilities that may serve as stepping stones for next exploration. Then the social partner suggests one if these stepping stone configurations as a goal. In this way the agent gradually discovers new configurations and extends its exploration space with only little help from the social partner.

•	The realization and implementation of the model seems to be solid and consists of a graph neural networks to represent the objects in the environment and their relations, and a knowledge graph to store previously seen semantic configurations.

**Summary Of The Paper:**

The paper targets goal-oriented reinforcement learning tasks with possible intervention of a virtual social partner assistant. It presents experiments for exploring object manipulation with 5 different blocks, in which an agent discovers configurations based on its own intuitive exploration, as well as guidance for other configurations from which it can further explore. The mount of intervention by the social partner is varied (from 0% to 100%), showing that 2-20% are sufficient for significant improvement over exploration without social partner.

**Summary Of The Review:**

Interesting approach based on the notion of Zone of Proximal Development for modeling goal-oriented reinforcement learning with social partner assistant

---

> ### Author Response · Authors · 2021-11-17
> **Answer to Reviewer 1**
>
> We thank Reviewer 1 for their feedback about the manuscript. Here we present the additional features we included in the new framing of the paper and that are specific to Reviewer 1's appreciations. More general details about the new structure of paper are addressed in detail in the general response.
>
> __The HME protocol: Benefiting social interaction within an autotelic agent__
>
> Reviewer 1 (R1) appreciated the novelty introduced by the HME architecture. We thank R1 for that and present additional details introduced in the new version and that we think could be of interest. Namely, we reinforced the coupling of individual autotelic learning and social interventions with an __internalization mechanism__. Using the latter, the autotelic agent can __rehearse__ the social goal proposals even when the SP is not present. Concretely, we keep track of the pair of goals that the SP proposes only if the agent fails to reach the goal that is beyond its frontier of knowledge. Hence, during autotelic episodes, the agent could replay the goal proposals for which it has failed. We believe this better highlights our contributions: coupling social interventions and autotelic learning influences the agent's intrinsic motivations, helps it explore more efficiently and breaks away from _physical situatedness_ during autotelic episodes.
>
> __Gradual discovery of new configurations and extending exploration space__
>
> R1 appreciated how social interventions help the agent explore the space of goals that are out-of-reach due to the physical situatedness and the agent's current policy limits. We thank R1 for that. We further optimized the SP proposal strategy in the new version of the paper. In fact, in the initial version, we defined the frontier of the agent's knowledge as a set containing both the stepping stones and the terminal nodes (which are not stepping stones but just complex configurations that are farther from the root). This implies that when the SP proposes a terminal goal, no corresponding “beyond” goal can be found. In the experimental studies introduced in the new version of the paper, we use a more efficient strategy: the SP only proposes stepping stones at first. Together with the internalization mechanism introduced above, we found that this accelerates the spanning of the goal space. Besides, the role of the SP ends when there are no more stepping stones left, which happens relatively early (depending on the proportion of social interventions). This further limits the social intervention effort: once there are no stepping stones left, the SP can leave the agent on its own. the latter would continue learning in an autotelic fashion with the possibility of rehearsing the SP interventions.
>
> __Conclusion__
>
> We thank R1 for their appreciation of several aspects of our work, especially related to the novelty introduced by the HME protocol. In the new version, we introduced several algorithmic improvements which we think might be of interest to R1.

---

### Author Response · Authors · 2021-11-17
**General Response to all Reviewers (3/3)**

Following the new framing of the paper, we introduce the following baselines:

* _No Internalization Baselines_: For each proportion of social interventions presented in this paper, we define a baseline where the internalization mechanism is omitted. Our aim is to study the importance of taking into consideration the SP influence on the agent's intrinsic motivations.

* _ACL Baselines_: To investigate whether ACL could replace Social Interventions, we introduce two baselines: 1) The Learning Progress baseline (LP) (Colas et al. 2018) which implements an automatic curriculum learning based on the estimation of the agent's learning progress over classes of semantic goals. Concretely, we use the same classes as in evaluation. Before the training kicks off, an empty bucket corresponding to each class is predefined. As GANGSTR encounters new configurations, it stores them in the appropriate bucket. When sampling a goal, GANGSTR uses its LP estimate  to compute a probability distribution over buckets which favors sampling from the bucket with the highest LP; 2) The Value Disagreement Sampling baseline (VDS) (Zhang et al. 2020) which uses a goal proposal module that prioritizes goals maximizing the epistemic uncertainty of the Q-function. Concretely, we simultaneously train several neural networks to estimate the values of goals. To pursue a goal, the agent selects configurations proportional to the standard deviation of the trained networks.

* _No Semantic Graph Baselines_: For each proportion of social interventions presented in this paper, we define a baseline that does not use a semantic graph to determine the path of sub-goals leading to a particular final goal. Instead, the policy directly targets the sampled configuration.

Baselines 1 are used in Section 4.1, Baselines 2 are used in Section 4.2 and Baselines 3 are used in the ablative studies of Section 4.3.

__Conclusion__

We believe that the main concerns of the reviewers were about the contributions and some other minor issues. Based on their comments and suggestions, we believe the new organization and additional results have a lot improved the paper. We hope reviewers will find the time to read the updated version of the manuscript, that includes all the points discussed above and presents a clearer organization.

---

### Author Response · Authors · 2021-11-17
**General Response to all Reviewers (2/3)**

__The second contribution__ is the GANGSTR learning algorithm, a particular instance of the artificial autotelic agent required to use the HME interaction protocol. This particular instance is designed to address a manipulation domain. We argue that __efficient exploration__ and __light social interactions__ result from the HME protocol given the following sufficient conditions: 1) a semantic representation that defines a structured high-level discrete goal space; 2) an internal graph-based representation of this goal space allowing the agent to decompose semantic goals intro sequences of intermediate sub-goals; 3) a goal-conditioned RL algorithm that can reach semantic goal configurations. More specifically, we consider a 5-object manipulation setup and two semantic predicates "close" and "above". This yields a discrete goal space of tens of thousands of possible semantic goal configurations, where each configuration is a vector of dimension 30 (10 for the “close” predicate and 20 for the “ above” predicate, and where each dimension corresponds to a predicate evaluated on a particular ordered pair of objects). As this goal space contains a vast diversity of configurations with different complexities, we believe it is sufficiently representative of the more general class of problems we aim to tackle, making it a good candidate as a proof-of-concept of the efficiency of our HME interaction protocol. First, we show that the slightest intervention of SP (with only 0.1% of social episodes) significantly improves the performance of GANGSTR. Second, we show that only 0.2% of social episodes is enough to master all the evaluation semantic configurations including stacks of 2 to 5 objects, pyramids of 3 objects, combinations of stacks and combinations of stacks and pyramids. Finally, we show that coupling social goal proposals with autotelic goal generation via our internalization mechanism yields better results than performing either of them separately (answering R2).

We insist on the fact that the implementation of the HME protocol using GANGSTR presented in this paper is one instance over many other possibilities, and future HME implementations may benefit from improvements such as curriculum-based generative modelling.

__Paper Organization__

The paper is now reorganized to better highlight the focus of our work by separating the main conceptual contributions from the secondary ones (answering R2). The new organization concerns both the methods and the experimental sections.

__The methods section__ first presents the general HME interaction protocol, the environment, the implementation of the GANGSTR algorithm—a particular autotelic agent designed for manipulation domains and subject to HME interactions—, and how HME's social learning is combined with GANGSTR's autotelic learning. Compared to the initial submitted manuscript, this implies reordering Sections 3.1, 3.2 and 3.3 and allocating more focus on the HME interaction protocol as a main conceptual contribution. The main algorithmic changes applied to GANGSTR are: 1) When proposing a goal in the frontier, the SP only considers stepping stones, rather than both stepping stones and terminal nodes, 2) after generating a path of intermediate configurations using its graph, GANGSTR achieves each sub-goal without exploration noise and 3) an internalization mechanism where GANGSTR keeps track of the pair of goals given by the SP if it fails reaching the “beyond” goal.

__The experimental section__ is reorganized in three sections to answer the following questions: 1) How does HME affect skill acquisition in GANGSTR? 2) Can Automatic Curriculum Learning (ACL) replace Social Interventions? and 3) How important is the decomposition process provided by GANGSTR’s semantic graph? In 1), we specifically investigate the global performance metrics, the per classe performance, the learning trajectories and the role of the internalization mechanism.

---

### Author Response · Authors · 2021-11-17
**Response to all Reviewers (1/3)**

_We sincerely thank all reviewers for their very useful feedback._

All the reviewers acknowledged the relevance of our research towards teachable autotelic agents of interest (R1, R2, R3). R1 found the claims and statements of the paper well-supported and correct. R3 considered our research question innovative but pointed to minor issues in some of the paper claims and asked for small changes to improve them. R2 questioned the lack of depth in the evaluation of our approach and the generality of our contributions. We reorganized the paper to answer all the major concerns listed above about the presentation and contributions of the manuscript and corrected the aforementioned minor issues. This general response re-expresses the problem we aim to tackle, clearly states our contributions towards its resolution and describes the reorganization of the methods and experimental sections we performed to better support our claims.

__Focus of the paper__

This paper argues that coupling social interventions with autotelic learning —that is learning by autonomously generating and pursuing goals according to intrinsic motivations—, facilitates the acquisition of a growing repertoire of skills. More specifically, we show how social signals can influence the learners' intrinsic motivations even with minimalistic interventions from the social partner. With social interventions, autotelic agents can better explore their complex goal space, breaking away from the constraints of their physical situatedness, while requiring very low effort from expert social partners.

__Contributions__

__The main contribution__ of the paper is an interaction protocol called Help Me Explore (HME) which is designed as a general solution for the problem of _physical situatedness_ in autotelic agents. With this protocol, social partners (SP) can foster exploration __beyond sensorimotor constraints__ through __only__ goal proposals —social partners do not have to provide demonstrations nor interact with the environment in any way—. To kick off, HME requires three conditions: 1) a goal-based environment stated in the form of a Markov Decision Process (MDP); 2) a social partner (SP) with a high-level structured representation of the environment's goal space and a model of the agent's current exploration limits; 3) an artificial autotelic agent capable of achieving high-level goals. In this paper, we consider semantic goals based on spatial predicates as high-level goals. Furthermore, we focus on semantic graphs as promising candidates for the SP's structured semantic goal space representation.

* _During social episodes_, a SP assesses the agent's knowledge of the goal space and proposes a first goal (G1) at its frontier (a stepping stone already discovered by the agent). The agent targets this goal using its current goal-conditioned policy. If G1 is reached, the SP proposes an adjacent second goal (G2) beyond the frontier. If the agent fails to reach G2 from G1, it internalizes the pair (G1, G2) so that it can rehearse the SP intervention during autotelic episodes.

* _During autotelic episodes_, the agent can either 1) uniformly select a goal from an initially empty buffer containing the set of semantic configurations it has discovered so far or 2) pick a pair from the list of pairs (G1, G2) that SP has proposed during social episodes and where the agent has failed to reach G2. The choice between both strategies is controlled by a hyper-parameter. With this second mechanism, the agent is not only able to train and master socially suggested goals that it has never seen, but can efficiently take itself to its Zone of Proximal Development (ZPD) during autotelic episodes. (i.e. when the SP is absent). We call this mechanism the _internalization mechanism_.

The choice of the nature of an upcoming episode (either social or autotelic) is controlled by an additional hyper-parameter indicating the proportion of social episodes. We vary this hyper-parameter from 0% (no social episodes) to 100% (only social episodes).

---

### Author Response · Authors · 2021-11-24
**Follow-up on Rebuttal**

We hope that reviewers have found time to read our response and the new version of our paper, and we are ready to answer their further questions if any.

---

### Decision · Program_Chairs · 2022-01-20

**Decision:**

Reject

**Comment:**

The paper introduces GANGSTR, an agent that performs goal-directed exploration both individually and "socially", with suggestions from a partner. It builds a graph of different configurations of a 5-block manipulation domain, and navigates this graph. The theoretical motivations for this algorithm are solid, and the direction is interesting. However, the results are less than convincing. In particular, as was mentioned in the discussion, it is not clear how this algorithm would generalize beyond the very simple 5-block manipulation domain. While having a simple benchmark has the advantage that you can explore it in depth, it also might obscure problems with the algorithm, unless complemented by a set of other benchmarks. It therefore seems that the paper is not ready for publication yet.